# The Caspase Homologues in Scallop *Chlamys farreri* and Their Expression Responses to Toxic Dinoflagellates Exposure

**DOI:** 10.3390/toxins14020108

**Published:** 2022-01-31

**Authors:** Zhongcheng Wei, Wei Ding, Moli Li, Jiaoxia Shi, Huizhen Wang, Yangrui Wang, Yubo Li, Yiqiang Xu, Jingjie Hu, Zhenmin Bao, Xiaoli Hu

**Affiliations:** 1MOE Key Laboratory of Marine Genetics and Breeding, College of Marine Life Sciences, Ocean University of China, Qingdao 266003, China; weizhongcheng@stu.ouc.edu.cn (Z.W.); dingwei@ouc.edu.cn (W.D.); lml940520@163.com (M.L.); shijiaoxia@ouc.edu.cn (J.S.); yangrui1121@126.com (Y.W.); liyubo@stu.ouc.edu.cn (Y.L.); 21200631119@stu.ouc.edu.cn (Y.X.); hujingjie@ouc.edu.cn (J.H.); zmbao@ouc.edu.cn (Z.B.); 2Laboratory for Marine Fisheries Science and Food Production Processes, Qingdao National Laboratory for Marine Science and Technology, Qingdao 266237, China; 3Laboratory of Tropical Marine Germplasm Resources and Breeding Engineering, Sanya Oceanographic Institution, Ocean University of China, Sanya 572000, China

**Keywords:** caspase, development, paralytic shellfish toxin, Zhikong scallop, *Chlamys farreri*

## Abstract

The cysteine aspartic acid-specific protease (caspase) family is distributed across vertebrates and invertebrates, and its members are involved in apoptosis and response to cellular stress. The Zhikong scallop (*Chlamys farreri*) is a bivalve mollusc that is well adapted to complex marine environments, yet the diversity of caspase homologues and their expression patterns in the Zhikong scallop remain largely unknown. Here, we identified 30 caspase homologues in the genome of the Zhikong scallop and analysed their expression dynamics during all developmental stages and following exposure to paralytic shellfish toxins (PSTs). The 30 caspase homologues were classified as initiators (caspases-2/9 and caspases-8/10) or executioners (caspases-3/6/7 and caspases-3/6/7-like) and displayed increased copy numbers compared to those in vertebrates. Almost all of the caspase-2/9 genes were highly expressed throughout all developmental stages from zygote to juvenile, and their expression in the digestive gland and kidney was slightly influenced by PSTs. The caspase-8/10 genes were highly expressed in the digestive gland and kidney, while PSTs inhibited their expression in these two organs. After exposure to different *Alexandrium* PST-producing algae (AM-1 and ACDH), the number of significantly up-regulated caspase homologues in the digestive gland increased with the toxicity level of PST derivatives, which might be due to the higher toxicity of GTXs produced by AM-1 compared to the N-sulphocarbamoyl analogues produced by ACDH. However, the effect of these two PST-producing algae strains on caspase expression in the kidney seemed to be stronger, possibly because the PST derivatives were transformed into highly toxic compounds in scallop kidney, and suggested an organ-dependent response to PSTs. These results indicate the dedicated control of caspase gene expression and highlight their contribution to PSTs in *C. farreri*. This work provides a further understanding of the role of caspase homologues in the Zhikong scallop and can guide future studies focussing on the role of caspases and their interactions with PSTs.

## 1. Introduction

Apoptosis regulates a broad range of cellular processes, including stress responses, cell death, growth, and development [1,2,3,4]. Caspases (cysteine aspartic acid-specific proteases) are conserved intracellular cysteine-dependent proteases that regulate apoptosis [1,2]. After being activated upon oligomerisation or cleavage at specific aspartate residues, caspases initiate and execute programmed cell death [3]. Based on the functional and structural differences of caspase family members, mammalian caspases are classified into four groups: initiators (caspases-2/8/9/10), executioners (caspases-3/6/7), inflammatory caspases (caspases-1, 4, 5, 11, 12, and 13) [4], and keratinisation-related caspases (caspase-14) [5]. After sensing apoptotic signals, initiator caspases activate executioner caspases to degrade key structural proteins and activate downstream enzymes [6,7]. Inflammatory caspases regulate cytokine maturation during inflammation [4]. Keratinisation-related caspases have only been reported in vertebrates, where they function in epidermal keratinocyte differentiation [5]. All caspases have a conserved CASc domain that consists of a 5-peptide site consisting of Gln-Ala-Cys-x-Gly (QACxG) (X is R, G, or Q) that is responsible for enzyme catalytic activity [7]. A number of caspase families have been reported in vertebrates, and several studies have explored caspases in invertebrates in the past several decades. However, the study of invertebrate caspases has been confined to model species such as the roundworm (*Caenorhabditis elegans*) [8] and fruit fly (*Drosophila melanogaster*) [8,9].

The Mollusca phylum is the second most diverse animal group [10]. Within Mollusca, members of the bivalve order (including oysters, mussels, scallops, clams, etc.) are susceptible to toxins in the surrounding environments, and they require a strong apoptotic process to ensure body homeostasis [11,12]. Caspases in bivalves (such as clams, mussels, and oysters [13,14,15,16,17,18]) are assumed to be involved in apoptotic processes driven by pathogen infections [13], environmental stressors [13,19], and cell development [15,20]. For example, caspase-8 homologues in mussels may play a significant role in response to temperature stress and setting cellular thermal tolerance limits [18]. Both caspase-1 and caspase-3 in oysters act in several tissues or organs that degenerate after larvae settlement [20]. Paralytic shellfish toxins (PSTs) are one of the most toxic marine neurotoxins produced by dinoflagellates such as *Alexandrium* and *Gymnodinium* [21] and cause nerve paralysis by inhibiting sodium channels [22,23]. When exposed to *Alexandrium catenella*, PSTs trigger apoptosis in the cells of the mollusc oyster *Crassostrea gigas* [24,25,26]. Moreover, incubation with PSTs from microalgae (STX and GTX-2/3) causes haemocyte death with apoptosis hallmarks in *C. gigas* to activate the expression of caspases [26]. During these biological processes, caspases may function in a tissue-specific manner. For instance, after exposure to *A. catenella*, caspase-3/7 is overexpressed in haemocytes [24], while its activity decreases in the digestive gland [27]. In addition, various pure PST compounds can directly affect caspase activity [28,29]. For instance, the activities of caspase-3 and caspase-8 are increased after injection of STX into the mussel *Mytilus chilensis* [29] or incubation of GTX2/3 with the scallop *Nodipecten subnodosus* [28].

The Zhikong scallop (*Chlamys farreri*) (Jones et Preston, 1904) is distributed along the seacoasts of Russia, Korea, and China and is one of the main mariculture species in China. It is semi-sessile and epibenthic and usually attaches itself to rocks and detaches under adverse conditions [30]. *C. farreri* has been widely reported to be contaminated by PST toxins [31]. It has a strong ability to accumulate PSTs (up to 40,241 μg saxitoxin (STX) eq. per 100 g) and thus is frequently used for studying PST accumulation and toxicity [32]. The role of caspases in the digestive gland and kidney of *C. farreri* remains largely unknown, although the digestive gland often accumulates the highest concentration of incoming toxins [33], and the kidney transforms them [30,34]. In this study, we systematically studied the diversity of caspase family genes in *C. farreri* based on genome sequencing. We also explored the expression of caspase homologues during development and after exposure to the PST-producing dinoflagellates *A. minutum* and *A. catenella*.

## 2. Results and Discussion

### 2.1. Phylogeny, Diversity, and Copy Numbers of Caspase Homologues in C. farreri

A total of 30 caspase homologues containing the CAS domain were identified through genome-wide screening of *C. farreri*. To confirm the classification of the caspases and investigate their relationships with known caspases, a phylogenetic tree was built using the caspases in *C. farreri* and other reference caspases documented in public databases. Based on the tree topology, 6 caspases in *C. farreri* were perfectly clustered with previously reported caspases-8/10 (labelled blue), 6 clustered with caspases-2/9 (labelled purple), 16 clustered with caspases-3/6/7 (labelled orange), and 2 clustered with caspase-3/6/7-like genes (labelled black) (Figure 1). An exception was the relationship with caspase-3/6/7-like genes which formed a relatively distant branches with caspases-3/6/7, but were closer to the vertebrate caspase-14 branch (Figure 1). These results suggest divergent evolution of the caspase-3/6/7 sub-group between invertebrates and vertebrates. Low support values (less than 30%, not shown in the tree) may be due to ambiguous sequence alignment caused by the high diversity of caspase homologues (different caspase gene categories spanning vertebrates and invertebrates). The caspase-2/9 and caspase-8/10 genes belong to the initiator caspase category, and caspases-3/6/7 belong to the executioner caspase category.

*C. farreri* possessed these two categories of caspases, whereas the other two caspase categories, namely, inflammatory caspases (caspases-1, 4, 5, 11, 12, 13) and keratinisation-related caspases (caspase-14), were not detected (Table 1). The findings were similar for *C. gigas*, a close relative of *C. farreri*, and the African clawed frog *Xenopus laevis* (Table 1). In contrast, mouse (*Mus musculus*) and human (*Homo sapiens*) possessed inflammatory caspases and the keratinisation-related caspase (Table 1). Inflammatory caspases are known to have important roles in the innate immune response, where they induce pyroptosis to halt the intracellular replication of pathogens and release pro-inflammatory signals [35]. To date, the keratinisation-related caspase, caspase-14, has only been identified in vertebrates, so it is likely to be related to epidermal differentiation in vertebrates [36]. In humans, caspase-14 encodes a protease that functions in epidermal keratinisation, which is important for periodontal health [37]. The absence of inflammatory caspases and keratinisation-related caspases in *C. farreri* suggests that the role of caspases in bivalves is not as important as that in vertebrates.

However, compared to vertebrates, caspases-2/9, caspases-8/10, and caspases-3/6/7 in *C. farreri* have higher copy numbers (Table 1). Specifically, no more than 14 caspases were found in the genomes of vertebrates, whereas 30 caspase homologues (12 initiators and 18 executioners) were identified in the *C. farreri* genome. Similarly, 40 caspase homologues (12 initiators and 28 executioners) were found in the *C. gigas* genome. These findings suggest a bivalve-specific expansion of caspase homologues, specifically initiators and executioners, in comparison to vertebrates. It has been demonstrated that the expansion of caspases in the oyster *C. gigas* may play roles in apoptosis, the immune response, and development [38]. We hypothesise that the caspases in *C. farreri* may have similar functional roles. 

### 2.2. The Protein Structure of Caspases in C. farreri

To study the protein structural characteristics of caspases in *C. farreri*, the intron–exon organisation was profiled and is shown in Appendix A. The number of exons ranged from 3 to 17 (Appendix A), indicating significant variation in the exon–intron organisation patterns. The lengths of the caspase ORFs ranged from 834 to 2292 bp and, accordingly, encoded proteins with 277 to 763 amino acids. The molecular weights (MW) of these caspase proteins varied from 31.37 to 86.01 kDa with predicted isoelectric points between 4.65 and 8.85.

The domain distribution patterns of caspases in *C. farreri* are shown in Figure 2. All caspases possessed a CASc domain, whereas the presence or absence of N-terminal CARD, DED, and DD domains could classify these caspases into different groups (Figure 2). Almost all of the executioner caspases (caspases-3/6/7 and caspases-3/6/7-like) contained a conserved CASc domain. Similar to caspases in vertebrates, six caspase-2/9 proteins contained CARD and CASc domains. Four caspase-8/10 proteins contained DED and CASc domains. One caspase-8/10 protein contained a DD domain and a CASc domain, and one caspase-8/10 protein contained only a CASc domain (Figure 2). 

Sequence alignment of CASc domains revealed a relatively conserved 5-peptide site (QACxG) that did not contain alanine, cysteine, or glycine in many sequences (Appendix A). The variation in the functional domains amongst caspases has also been previously discovered [39]. The distribution patterns of CARD, DED, and DD domains (Figure 2) were roughly consistent with the phylogenetic relationship of the caspase homologues (Figure 1), in which the initiator caspase homologues were separated from executioner genes. The CARD domain is essential for the binding ability to Cyt-c and Apaf-1 proteins, which is required for the downstream activation of executioners [40]. In vertebrates, the DED domain is necessary for the proximity-induced activation of proenzymes [8], and the DD domain is a homotypic domain with a similar function to that of the DED domain [41]. In this study, the presence or absence of the DD and DED domains in caspases-8/10 implied gene reduction or duplication events. The significance of the absence of the DD and DED domains in regulating the binding ability of caspases in *C. farreri* requires further study. The CASc domain is essential for the catalytic activity of caspases [7]. The variation of amino acids in the QACxG sites also implies conserved and specific biological activities of caspases in *C. farreri.*

### 2.3. Spatiotemporal Expression of Caspase Homologues in C. farreri

To further explore the potential function of caspase homologues in *C. farreri*, the temporal expression dynamics of these genes were investigated during embryonic and larval development. Overall, most caspase-8/10 homologues showed a low level of expression, whereas nearly all of the caspase-2/9 homologues analysed were expressed at high levels during the different stages of development (Figure 3A). In contrast, many of the caspase-3/6/7 homologues showed differential expression patterns between earlier and late stages of development (Figure 3A). For instance, CF.60899.92-caspase-3/6/7, CF.41779.40-caspase-3/6/7, and CF.4213.1-caspase-3/6/7 were highly expressed from the D-stage veliger to the juvenile stage and were expressed at very low levels from zygote to trochophore. The CF.47297.24-caspase-3/6/7 was only silenced from the zygote to the 2–8-cell stage. Five caspase-3/6/7 homologues (CF.61727.3, CF.24861.4, CF.61175.1, CF.25603.9, and CF.61415.1) and two caspase-3/6/7-like homologues were silenced throughout all stages of development (Figure 3A). In addition, CF.11121.7-caspase-3/6/7 was expressed at low expression levels throughout all stages of development (Figure 3A). 

The caspase-2 homologues in bivalves were found to act as initiators of programmed cell death [38]. They are expressed in almost all metazoan cells and are activated in response to intrinsic and extrinsic apoptotic signals [42]. The caspase-9 homologues in *Drosophila* regulate programmed cell death and exert control on organ degeneration and during early metamorphosis, which marks the transition from larval to juvenile life stages [43,44]. These findings are consistent with previous results indicating that caspase-2/9 homologues play roles in all developmental stages. The inhibition of caspase-3 activity completely abolishes metamorphosis of the Cnidaria *Hydractinia echinata* [43,44], suggesting a role for caspase-3 in metamorphosis. However, not all caspase-3 homologues are likely to function in the metamorphosis *of C. farreri*, considering the high copy numbers of this gene and its specific expression in certain stages of development.

Spatial gene expression analysis of caspases in different organs at the adult stage (Figure 3B) showed that five of the six caspase-2/9 homologues were expressed in all organs. Four of the six caspase-8/10 homologues displayed organ-dependent expression, with higher expression levels in the digestive gland and gills. Compared to the earlier stages, most of the caspase-3/6/7 homologues were expressed at high levels during the adult stage in all organs (Figure 3B). Three caspase-3/6/7 homologues (CF.61727.3, CF.24861.4, and CF.61175.1) were repressed in different organs (Figure 3B). Taken together, the caspase-2/9 genes tend to be expressed throughout all stages and in all organs of the adult. These data suggest that these genes may play key roles in the development of *C. farreri*. Many caspase-8/10 homologues are only expressed in certain organs in the adult stage. These observations suggest that they may contribute to the differentiation of these organs. Nearly all of the caspase-3/6/7 genes displayed stage- or organ-dependent expression, suggesting that their expression is tightly regulated by intracellular signals. The silencing of caspase-3/6/7-like homologues in most of the stages of development and organs suggests a limited function during the development of *C. farreri*.

### 2.4. The Expression Patterns of Caspase Homologues in the Digestive Gland and Kidney after Exposure to Toxic Dinoflagellates 

*Alexandrium* strains produce different PST derivatives [45]. *A. minutum* AM-1, for example, mainly produces gonyautoxins (GTXs) with high toxicity, while *A. catenella* ACDH mainly produces N-sulphocarbamoyl derivatives (C1/2) with low toxicity [46]. The activities of caspase proteins in bivalve haemocytes have been found to be triggered by PST exposure [24,25,26]. Compared to haemocytes, the digestive gland accumulates higher concentrations of toxins after *C. farreri* is exposed to the PST-producing algae *Alexandrium*, and the kidney is responsible for toxin transformation from less to more toxic derivatives, mainly saxitoxin (STX) and neosaxitoxin (NeoSTX) [30]. This motivated us to perform a time-series analysis of caspase homologue expression in the digestive gland and kidney of *C. farreri* before and after exposure to the PST-producing dinoflagellates *A. minutum* AM-1 and *A. catenella* ACDH. The relative expression levels (indicated by fold-change values in comparison to the untreated organ) of caspase homologues in the digestive gland and kidney of *C. farreri* after exposure to the dinoflagellates are shown in Figure 4 and Figure 5, respectively. In the digestive gland, nine and seven caspase homologues were significantly (|log_2_FC| > 1 and *p*-value < 0.05) differentially expressed after exposure to AM-1 (Figure 4A) and ACDH (Figure 4B), respectively. The expression level of caspase homologues increased with the toxicity level of PST derivatives. For instance, four caspase-3/6/7 homologues (CF.61727.3, CF.60899.92, CF.24861.4, and CF.27679.13) were significantly up-regulated after AM-1 exposure, but only CF.61727.3-caspases-3/6/7 was significantly up-regulated after ACDH exposure (Figure 4). This might be due to the higher toxicity of GTXs produced by AM-1 compared to N-sulphocarbamoyl derivatives produced by ACDH. Moreover, the up-regulation of CF.61727.3-caspase-3/6/7 was acute and inductive, as it was almost unexpressed in the untreated digestive gland.

In the kidney, six and seven caspase homologues were significantly differentially expressed after exposure to AM-1 (Figure 5A) and ACDH (Figure 5B), respectively. However, unlike that in the digestive gland, the effect of these two PST-producing algae strains (AM-1 and ACDH) on caspase expression in the kidney seemed to be stronger, as the number of up-regulated (|log_2_FC| > 1) caspase homologues in the kidney (12 and 6) was greater than that in the digestive gland (7 and 4). These results suggested that many more caspase homologues were expressed in response to the higher toxicity in the kidney. In parallel, three and two caspase homologues were significantly up-regulated in the kidney after AM-1 and ACDH exposure, respectively. This can be explained by the fact that PST derivatives, regardless of their types, will be converted from low to high toxicity in the scallop kidney. Notably, CF.61727.3-caspases-3/6/7 showed the highest expression level after 10 days of exposure, and its expression was acutely triggered by AM-1 but chronically induced by ACDH, suggesting that the toxicity level of PSTs might be related to the speed of response in the kidney. Furthermore, for certain caspase homologues, we observed opposite responses to the PST-producing algae in the digestive gland and in the kidney. For example, the CF.12119.2-caspase-3/6/7-like homologue was up-regulated in the digestive gland and down-regulated in the kidney by both algae strains (Figure 4 and Figure 5), also suggesting an organ-dependent response to PSTs.

## 3. Materials and Methods

### 3.1. Identification and Sequencing of Caspase Homologues in C. farreri

The genome and transcriptome data of *C. farreri* were downloaded from our previous studies [30]. De novo predicted protein-coding genes were aligned to the genome to support the mRNAs’ existence. Then, protein-coding genes were translated using the software ORF Finder [47]. The caspase protein sequences from vertebrates *Homo sapiens*, *Mus*
*musculus*, and *Xenopus laevis* and the invertebrate *C. gigas* were downloaded from the UniProt website (http://www.uniprot.org/, accessed on 10 September 2021) and subjected to blastp against all protein sequences of *C. farreri* to search for candidate caspase homologues (E-value < 1 × 10^−5^). Conserved caspase domains were identified by searching against the Pfam database [48] (PF00656: Peptidase_C14) based on the hidden Markov model (HMM) (E-value < 1 × 10^−5^). The functional domains were further confirmed using the tools SMART (Heidelberg, Germany) [49] and CDD (Bethesda, MD, USA) [50]. The caspase protein characteristics, such as protein lengths, molecular weights (kDa), isoelectric points (pI), instability indices (II), aliphatic indices, and average hydropathicity (GRAVY) values, were analysed using the ExPASy online server (Lausanne, Switzerland) [51].

### 3.2. Multiple Alignment and Phylogenetic Analysis

The CASc domains from *C. farreri* and other species, including vertebrates *H. sapiens, M. musculus,* and *X. laevis* and the invertebrate *C. gigas*, were used to perform phylogenetic analysis. The other caspase protein sequences were downloaded from the UniProt and NCBI databases and used as references. Multiple alignments of CASc domains derived from different species were conducted using the Clustal W program [52]. The best model for the phylogenetic tree was predicted using Prottest [53]. The maximum likelihood phylogenetic tree was constructed by using RAxML [54] with the WAG + G model and 1000 bootstrap pseudo-replicates. The phylogenetic tree was visualised by using iTOL (Heidelberg, Germany; Hinxton, England; Grenoble, France; Hamburg, Germany; Rome, Italy and Barcelona, Spain) [55].

### 3.3. Transcriptomic Analysis of Caspase Homologues during the Development of C. farreri 

Based on the published RNA-seq data of *C. farreri* [30], the expression of caspase homologues at different developmental stages was investigated. Embryo/larva samples at different stages of development were harvested. These stages included the zygote, 2–8 cells, blastula, gastrula, trochophore, D-stage larvae, early umbo larvae, middle umbo larvae, late umbo larvae, creeping larvae, and juvenile scallops. Different adult organs/tissues were also collected for RNA-seq, including mantles, gills, gonads, kidneys, digestive glands, striated muscles, smooth muscles, eyes, feet, and haemocytes. RNA-seq libraries were constructed using the NEBNext Ultra RNA Library Prep Kit for Illumina (NEB, Boston, United States) and sequenced using the Illumina platform. The expression of caspase homologues was normalised by transcripts per million mapped reads (TPMs). Each sample was analysed based on at least three biological replicates.

### 3.4. Transcriptomic Analysis of the Caspase Homologues in Response to PSTs

Strains of toxic dinoflagellate *Alexandrium* produce various PST derivatives [45]. *A. minutum* AM-1 mainly produces highly toxic gonyautoxin derivatives (e.g., GTX1, GTX2, GTX3, and GTX4) [56], whereas *A. catenella* ACDH mainly produces low toxic N-sulphocarbamoyl derivatives (C1 and C2) [34,46,57]. To explore the effect of different PST derivatives on *C. farreri*, two-year-old Zhikong scallops were domesticated in laboratory conditions for three weeks and then fed PST-producing dinoflagellates *A. minutum* AM-1 or *A. catenella* ACDH. The two *Alexandrium* dinoflagellates were provided by the Institute of Oceanology, Chinese Academy of Sciences. The algae cells were cultivated independently and then harvested at the late exponential growth phase. Before the exposure experiment, the PST profiles in the two *Alexandrium* dinoflagellates were measured by a high-performance liquid chromatography with tandem mass spectrometry (HPLC–MS/MS) analysis following the steps described in a previous study [58]. Each scallop was fed 2500 cells/mL/day of dinoflagellates. Scallops were sampled on day 0 (the control group without PST treatment) and on days 1, 3, 5, 10, and 15 (the experimental groups) after exposure to dinoflagellates. Samples at each time point included three replicates. RNA was extracted from the digestive glands and kidneys and followed by RNA library construction and data analysis, as mentioned above. The expression of caspases was normalised and represented by RPKM. The differential expression of the caspase homologues in each experimental group was indicated by log_2_ (fold change), namely, log_2_ (RPKM in the experimental group/RPKM in the control group). The edgeR [59] R package (version 4.0.2) was used to determine the differentially expressed genes using the cutoff |log_2_(fold change)| > 1 and *p*-value < 0.05. Heat maps were drawn using the pheatmap [60] R package.

## 4. Conclusions

We performed a comprehensive analysis of caspase homologues in *C. farreri*. In total, we identified 30 caspase homologues that were initiators (caspases-2/9 and caspases-8/10) and executioners (caspases-3/6/7 and caspases-3/6/7-like) and had higher copy numbers compared to caspases in vertebrate genomes, suggesting gene expansion. Integrating genomic and transcriptomic analyses indicated different expression patterns between caspase-2/9, caspase-8/10, caspase-3/6/7, and caspase-3/6/7-like genes during the development of *C. farreri* and in response to PSTs. Specifically, we demonstrated that caspase-2/9 genes tended to be expressed in all stages of development and all differentiated organs, whereas the higher expression of caspase-8/10 genes was only observed in the digestive gland and kidney in the adult stage. The number of significantly up-regulated caspase homologues in the digestive gland increased with the toxicity level of PST derivatives. This might be due to the higher toxicity of GTXs produced by AM-1 compared to the N-sulphocarbamoyl analogues produced by ACDH. However, the effect of these two PST-producing algae strains on caspase expression in the kidney seemed to be stronger, possibly because the PST derivatives were transformed into highly toxic compounds in the scallop kidney. Organ-dependent response to PSTs can be further supported by the opposite responses of certain caspase homologues in the digestive gland and kidney. Based on these findings, we propose that the expression of caspase homologues is specifically controlled in the development of *C. farreri* and in response to PST treatment, implying roles in apoptosis and responses to environmental stress. Our work lays the foundation for further functional exploration of particular caspase homologues in *C. farreri.*

## Figures and Tables

**Figure 1 toxins-14-00108-f001:**
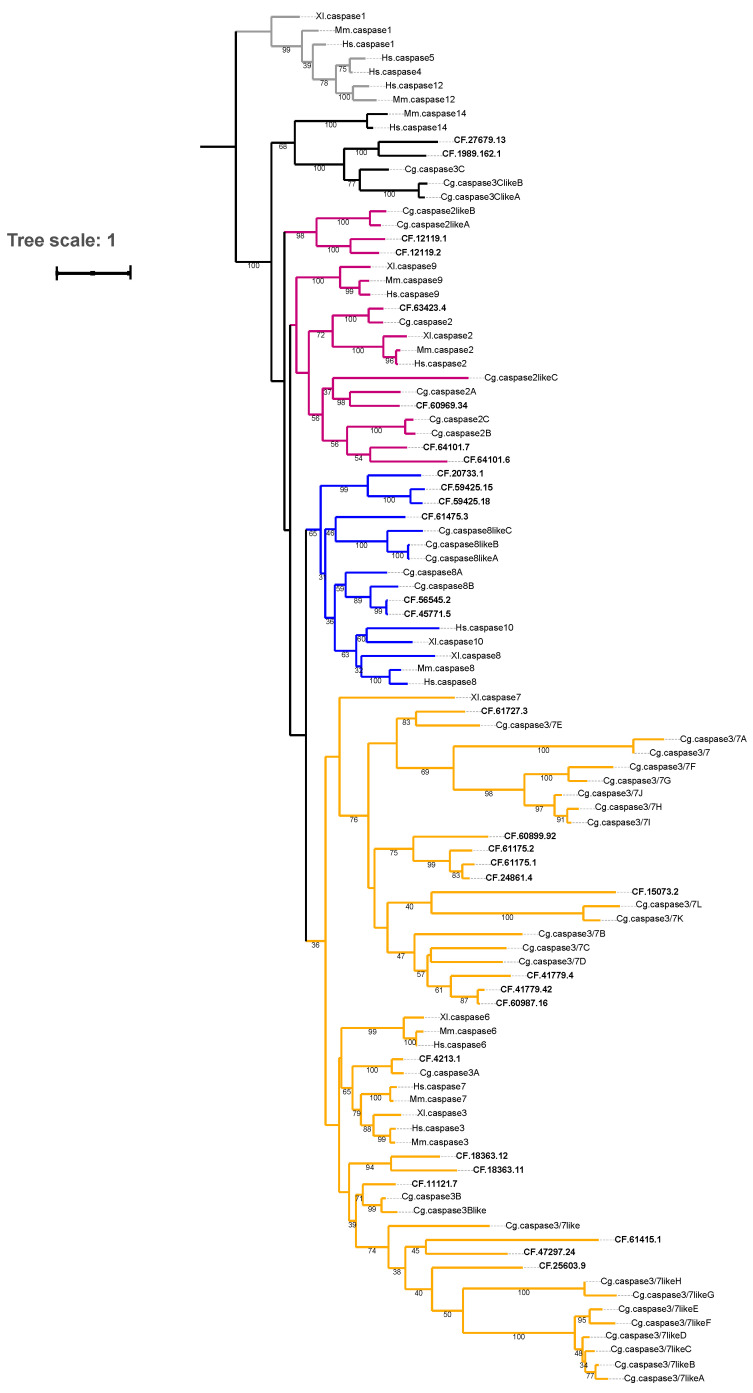
Phylogenetic analysis of caspase homologues from selected organisms. The tree is drawn to scale, with branch lengths measured based on the number of substitutions per site. Low support values (values less than 30%) are not shown. The caspase homologues identified from the *C. farreri* genome are highlighted in bold. Reference genes from other species were downloaded from public databases. Hs: *H. sapiens*, Mm: *M. musculus*, Xl: *X. laevis*, Cg: *C. gigas*, CF: *C. farreri*.

**Figure 2 toxins-14-00108-f002:**
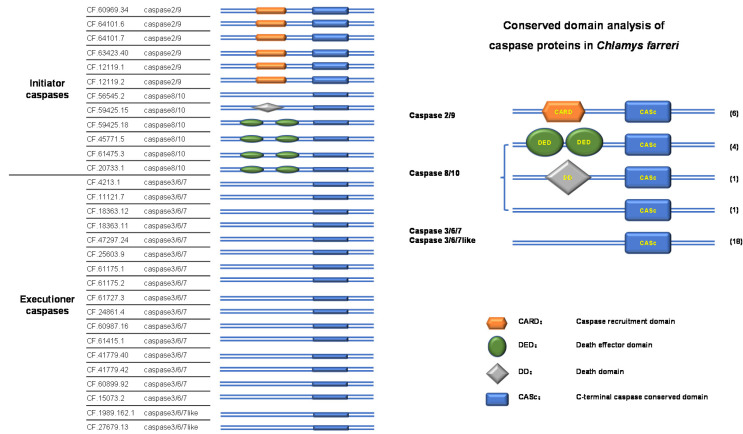
Conserved domains of the 30 caspase proteins in *C. farreri*. The five representative domain distribution patterns and corresponding gene numbers are given in the right panel.

**Figure 3 toxins-14-00108-f003:**
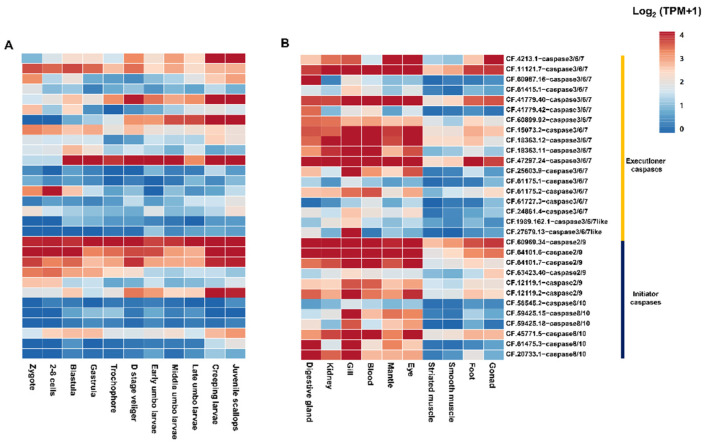
The expression profiles of caspase homologues in embryos (**A**) and adult organs/tissues (**B**) of *C. farreri*. The mRNA levels represented by log_2_ (TPM + 1) values are shown in the gradient heatmap with colours ranging from blue (low expression) to red (high expression).

**Figure 4 toxins-14-00108-f004:**
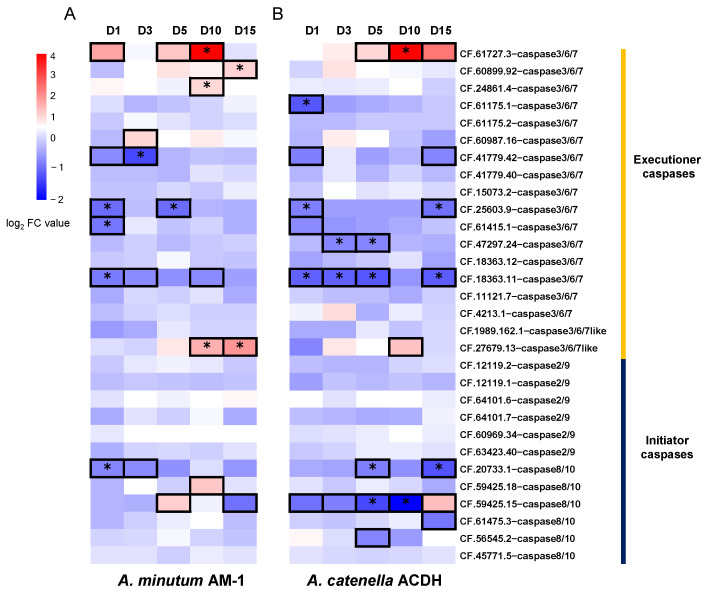
The expression profiles of caspase homologues in the digestive gland of *C. farreri* after exposure to PST-producing dinoflagellates (**A**) *A. minutum* (AM-1) and (**B**) *A.*
*catenella* (ACDH). The expression levels are indicated by the logarithm of fold change (log_2_FC) values compared to untreated organs. The cells with bold lines indicate that the expression levels are |log_2_FC| > 1, and the asterisks indicated a significantly different expression level with *p*-value < 0.05.

**Figure 5 toxins-14-00108-f005:**
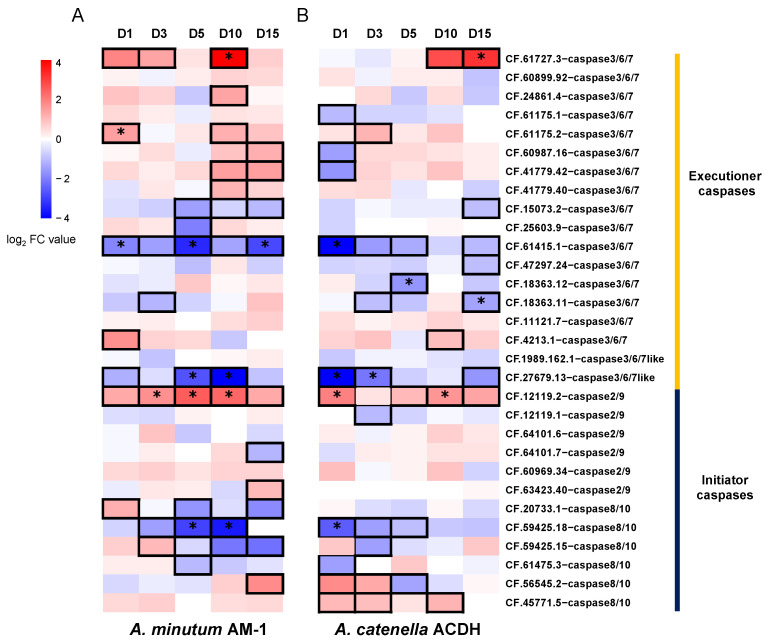
The expression profiles of caspase homologues in the kidney of *C. farreri* after exposure to PST-producing dinoflagellates (**A**) *A. minutum* (AM-1) and (**B**) *A.*
*catenella* (ACDH). The expression levels are indicated by the logarithm of fold change (log_2_FC) values compared to untreated organs. The cells with bold lines indicate that expression levels are |log_2_FC| > 1, and the asterisks indicate a significantly different expression level with *p*-value < 0.05.

**Table 1 toxins-14-00108-t001:** The distribution of caspase gene homologues in different species. The numbers of caspases in *C. farreri* are highlighted in bold.

Gene Category	Gene Name	*Mus musculus*	*Homo sapiens*	*Xenopus laevis*	*C. gigas*	*C. farreri*
Inflammatory caspases	caspase-1/4/5/11/12/13	5	3	0	0	**0**
Initiator caspases	caspase-2/9	2	2	2	7	**6**
	caspase-8/10	2	2	2	5	**6**
Executioner caspases	caspase-3/6/7	3	3	3	28	**18**
Keratinisation-related caspases	caspase-14	1	1	0	0	**0**
Total number		14	12	7	40	**30**

## Data Availability

The genomic and transcriptomic data in this study were deposited in NCBI BioProject (accession number PRJNA185465). Some unpublished data are included in our ongoing project and will be released soon; therefore, if any readers are interested in these data, they are welcome to contact the corresponding author for further query.

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
