# Peer review of "The Caspase Homologues in Scallop Chlamys farreri and Their Expression Responses to Toxic Dinoflagellates Exposure"

_toxins, 2022, doi:10.3390/toxins14020108_

Round 1

Reviewer 1 Report

This manuscript describes the identification of 30 caspase homologs in the genome of the clam Chlamys (Azumapecten) farreri, providing some insights into their regulation along development and following PST exposure. Overall, this is an interesting study, which provides an overview on the repertoire of these important genes in a bivalve species, trying to also give some insights into their potential involvement in PST response. There are, however, several point that might require improvement.

L45: “invertebrate caspases are confined” should be replaces by “the study of invertebrate caspases is confined”

L48: “include oyster, muscles, scallop, and the clam, etc.”: improve the English language, e.g. muscles should be mussels, scallops should be plural, etc.

L57: exposed -> exposure

L61: muscles -> mussels

The introduction should provide some information concerning the few studies carried out so far on bivalve caspases. These are not many, but the authors in my opinion should provide a brief overview of the knowledge available so far on this topic. Also, some of these information could be reported in the discussion.

Figure 1: the meaning of “tree scale” is unclear. The authors should clarify what the scale bar stands for (substitutions per site?). Also, use italics for species names in the caption.

The section describing the phylogenetic tree and the classification of caspases should discuss the few uncertainties (i.e. low support values) that characterize some basal nodes od the tree.

L104: this is an overstatement: inflammatory caspases are just responsible of a very small part of bivalve immune response, so stating that their absence suggests that the bivalve immune response, as a whole, is “not as complex as in vertebrates” is misleading. I suggest to modify this statement by rather writing sometime along the lines of “this suggests that in bivalves caspases do not play a role as important as in vertebrates in the regulation of inflammatory response”.

L121-123: this appears to be a range of size way too high to be reasonable, considering that the caspase domain itself is about 200aa long. How did the authors check for gene model completeness? Bivalve genome are always imperfect, with some regions suffering from long stretches of undetermined sequence (Ns) or misassemblies, and the annotation of gene models could be affected as well. The authors should therefore make sure that all caspase gene models are likely complete and report their uncertainties accordingly.

L125: “showed” -> shown

L144: “evolutionary approaches” is not an appropriate term. The authors should rather discuss the polyphyletic origin of these two groups of caspases.

Fig.3 caption: specify that Log(FPKM+1) have been plotted. Also, using TPMs instead of FPKMs would be more appropriate, as FPKMs do not allow a reliable comparison among samples.

L215: “exposed” -> exposure. Please correct this also elsewhere in the text

L256: the authors should explain that ORF finder was only used for transcriptome data. I guess a gene annotation track was already available for the genome.

L260: specify the ID of the Pfam HMM model used for this screening

L271-272: the authors claim that the “ML model” was used. However, Maximum likelihood is not, per se, a “model”, but rather a tree building algorithm. What was the molecular model of evolution suggested by modeltest in this case? This should be either a WAG, LG, GTR, etc.

L284-285: as mentioned above, to ensure better comparability among samples, TMPs should have been used instead.

Section 3.4: if this is new data, then the reads should be deposited in a public sequence repository (such as SRA) and the Bioproject accession ID should be provided.

Author Response

Dear Editor,

Thank you very much for your handling of our manuscript, and we sincerely appreciate the reviewer’s helpful suggestion to improve the quality of our manuscript. We have revised the manuscript according to the reviewers’ comments, with a point-by-point response being provided below. The modified parts have been marked-up in blue typeface in the revised version. Please see the attachment.

Reviewer 2 Report

The manuscript entitled “The caspase homologues in scallop Chlamys farreri and their expression responses to toxic dinoflagellates exposure” aimed to identify 30 caspase homologues in the genome of the C. farreri and analysed their expression dynamics during complete developmental stages and following exposure to paralytic shellfish toxins.

Please indicate the origin of different strains of toxic dinoflagellate Alexandrium. How was toxin production measured from these dinoflagellates?

Page 4, line 101 – related to epidermal, what?

Minor remarks:

Page 1, line 16 – please change to “After exposure to different…”

Page 4, line 96 – please change to “the human Homo sapiens…”

Author Response

Dear Editor,

Thank you very much for your handling of our manuscript, and we sincerely appreciate the reviewer’s helpful suggestion to improve the quality of our manuscript. We have revised the manuscript according to the reviewers’ comments, with a point-by-point response being provided below. The modified parts have been marked-up in blue typeface in the revised version.

Response to Reviewer 2 Comments

The manuscript entitled “The caspase homologues in scallop Chlamys farreri and their expression responses to toxic dinoflagellates exposure” aimed to identify 30 caspase homologues in the genome of the C. farreri and analysed their expression dynamics during complete developmental stages and following exposure to paralytic shellfish toxins.

Point 1: Please indicate the origin of different strains of toxic dinoflagellate Alexandrium. How was toxin production measured from these dinoflagellates?

Response 1: Thanks for your comments on this work. “The two Alexandrium dinoflagellate were provided by Institute of Oceanology, Chinese Academy of Sciences. The algae cells were cultivated independently and then harvested at the late exponential growth phase. Before the exposure experiment, the PST profiles in the two Alexandrium dinoflagellates were measured by a high-performance liquid chromatography with tandem mass spectrometry (HPLC-MS/MS) analysis following the steps described in a previous study [58]”. We have also added related information in the acknowledgements. Please see lines 340-346 and lines 392-395 in the marked-up manuscript.

Point 2: Page 4, line 101 – related to epidermal, what?

Response 2: Revised according to the comments. Please see the line 140 in the marked-up manuscript.

Minor remarks:

Point 3: Page 1, line 16 – please change to “After exposure to different…”

Response 3: Revised according to the comments. Please see line 37 in the marked-up manuscript.

Point 4: Page 4, line 96 – please change to “the human Homo sapiens…”

Response 4: Revised according to the comments. Please see lines 134-135 in the marked-up manuscript.
